# HYBRID CONTRASTIVE TRANSFORMER FOR VISUAL TRACKING

## ABSTRACT

Visual object tracking is a research hotspot in the field of computer vision, and has been widely applied in video surveillance, human-computer interaction, unmanned driving and other fields. At present, the object trackers based on Transformer have good performance, but they still face the challenge of confusing target and background in the feature extraction process. To address this issue, we propose a Hybrid Contrastive Transformer Tracker (HCTrack) in this paper, which combines contrastive learning to improve the ability of distinguishing the target and the background in video. Furthermore, a hybrid feature interaction module is presented to realize multi-level information exchange between the features of template and search regions and capture the target-related semantic information of the search frames comprehensively. Additionally, we design a redundant information pruning module to adaptively eliminate the redundant backgrounds according to the global scene information, thereby reducing the interference of the background to the target feature. HCTrack achieves superior tracking accuracy on the GOT-10k and TrackingNet datasets compared to other state-of-the-art trackers, while maintaining fast inference speed, as the contrastive learning strategy is only adopted during training model.

## 1 INTRODUCTION

Object tracking aims to locate the target in each frame of a continuous video sequence while providing scale and position information. It plays a pivotal role in understanding and analyzing various moving objects of videos, and has been widely applied in video surveillance, human-computer interaction, unmanned driving, and so on. However, tracking object in the real-world scenarios presents numerous challenges such as lighting variations, scale adjustments, motion blur, and interference from similar objects within complex scenes.

In recent years, there were a great many deep learning-based trackers, which could be broadly categorized into two groups: trackers based on Siamese network and trackers based on Transformer. SiamFC(Bertinetto et al., 2016) firstly integrated Siamese network into object tracking by utilizing a convolutional neural network (CNN) to extract the feature of the search frame and the template. SiamRPN++(Li et al., 2019a) and SiamDW(Zhang & Peng, 2019) enhanced the feature representation capability of model to improve the tracking performance. Subsequently, anchor-based(Fu et al., 2021a; Li et al., 2018; Zhu et al., 2018) trackers and anchor-free(Han et al., 2021; Xu et al., 2020; Zhang et al., 2020) trackers employed different target scale regression strategies to enhance adaptability to the scale change of target. In addition, some template update strategies(Fu et al., 2021b; Yang et al., 2023; Wang et al., 2023) were introduced to accommodate variations in the appearance of the target during tracking. However, due to the limited local receptive field of CNN, it is difficult to obtain global information of samples, so that the Siamese-based trackers are unable to accurately locate the targets in the complex scenarios such as occlusion, motion blur, and background interference.

In contrast, as the core module of Transformer, attention mechanism can effectively capture the global contextual information of video. Currently, the existing Transformer-based object tracking methods could be divided into two-stage and single-stage Transformer-based trackers. The two-stage Transformer-based trackers(Chen et al., 2021; Yan et al., 2021) still struggle to accurately extract the target- specific feature during the feature extraction, which could lead to a loss of some

effective detail information. The single-stage Transformer-based trackers(Chen et al., 2022; Song et al., 2023; Wu et al., 2023; Zhao et al., 2024) utilized a Transformer-based backbone to interact with the template feature and the features of search frames, and directly extract the target-related information of search frames in the process of extracting feature. However, the existing Transformer-based trackers tended to confuse the target with the background due to their premature interaction, which would result the misclassification.

Contrastive learning is a learning method by comparing the similarity between different samples, which can enhance the discriminability of model by distinguishing between the positive and negative samples. Similar to contrastive learning, object tracking requires maximizing the similarity among the same objects in different frames of the same video sequence and the difference among different objects. Therefore, we propose a Transformer-based tracker incorporating contrastive learning, called as a hybrid contrastive Transformer tracker (HCTrack), to guide the model to accurately track the objects in this paper. Firstly, we develop a contrastive learning strategy, which constructs the positive and negative sample pairs and utilizes their semantic label information to effectively improve the similarity among same-class targets and the dissimilarity among different-class targets. Furthermore, considering that most existing tracking methods using synchronous or asynchronous feature extractors only could extract limited target information, we present a hybrid feature interaction style involving a semantic self-association module and a cross-layer semantic association module (CSA). CSA associates the template features across multiple layers during the deep feature extraction of search frame to accurately capture the target-related semantic information. Additionally, in order to reduce the focus of the model on the background region, we devise a redundant information pruning (RIP) module, which adaptively prunes the redundant background region features based on the scene complexity. Additionally, in order to reduce the focus of the model on the background region, we devise a redundant information pruning (RIP) module, which adaptively prunes the redundant background region features based on the scene complexity. The experimental result of HCTrack on GOT-10k and TrackingNet datasets demonstrates that HCTrack has competitive performance in comparison to other state-of-the-art object tracking methods. The main contributions of this paper are as follows:

(1) In order to mitigate the confusion between the target and the background, we introduce the contrastive learning into the transformer-based tracker, and propose a hybrid contrastive transformer tracker (HCTrack). Our designed contrastive learning strategy can enhance the ability of model to discriminate the targets and backgrounds in the process of objects tracking by using the label information of the constructed positive and negative samples.

(2) A semantic self-association module and a cross-layer semantic association module are presented, which make full use of multi-level template features and guide model to accurately learn the target-related feature.

(3) A redundant information pruning module is established for pruning the redundant background information, thereby reducing on the impact of a large number of complex backgrounds on the target during tracking .

## 2 RELATED WORK

### 2.1 SIAMESE-BASED TRACKERS

Siamese network has received a lot of attention in the field of target tracking because of their strong scalability and dual advantages in speed and accuracy. After SiamFC (Bertinetto et al., 2016) introducing siamese network into the object tracking is proposed, numerous improved tracking methods have been derived from it. DSiame(Guo et al., 2017) handled changes of the target over time through online update, and inspired a series of approaches(Zhang et al., 2019; Li et al., 2019b; Zheng et al., 2023; Zhao et al., 2024) that utilized spatio-temporal relationship. SiamMDM(Yang et al., 2023) developed a dynamic template update module and a score-based model for predicting target motion trajectory. (Wang et al., 2023) proposed a dynamic template updating strategy based on spatio-temporal information and designed a tracking confidence network to decide whether to update.

Later, inspired by the object detection methods, SiamRPN(Li et al., 2018) introduced region proposal network(Girshick, 2015) into object tracking, improving the adaptability of model to scale

variations. On the basis of SiamRPN, Ren et al. proposed a classical structure Faster-RCNN(Ren et al., 2016) and Cai & Vasconcelos gave a cascading structure Cascade R-CNN(Cai & Vasconcelos, 2018). SiamRPN++(Li et al., 2019a) and SiamDW(Zhang & Peng, 2019) applied deep backbone networks in the object tracking framework to improve the model's ability to represent features more effectively. Meanwhile, some methods introduced anchor-free networks to adapt to the changing shape of the target. SiamFC++(Xu et al., 2020) removed anchor priors and utilized a quality assessment branch to refine the target localization results of the classification branch. Although the Siamese network-based object tracking methods perform well, it is difficult to obtain global information from video sequences due to the limited receptive field of CNN. Consequently, these methods face the challenges in accurately locating targets with significant appearance changes in complex scenarios such as occlusion, motion blur, and background interference.

## 2.2 TRANSFORMER-BASED TRACKERS

Inspired by the significant success of Transformer(Vaswani, 2017) architecture in various tasks(Dosovitskiy, 2020; Zhang et al., 2024; Zong et al., 2023; Venkataramanan et al., 2023), scholars began applying it in the field of object tracking. At present, the Transformer-based object tracking methods can be broadly categorized into two types: two-stage and single-stage Transformer-based tracking methods.

The two-stage Transformer-based object tracking methods mainly contains a feature extraction module and a feature fusion module. TransT(Chen et al., 2021) extracted the features of template and search frames in the feature extraction stage, and then interacted them by a Transformer structure in the feature interaction stage, thus capturing the target-related information in the search features. Yan et al. introduced a spatiotemporal Transformer network, named STRAK(Yan et al., 2021), which extended the traditional Transformer encoder-decoder framework to capture spatiotemporal features of video. Cao et al. proposed a Hierarchical Feature Transformer (HiFT)(Cao et al., 2021), which integrated multi-level features to efficiently learn the global dependency relationship among different levels of features. Videotrack(Xie et al., 2023) leverages the self-attention mechanism to simultaneously process both the spatial and temporal dimensions of videos, effectively enhancing its ability to capture features over long sequences. Afterwards, some two-stage Transformer-based object tracking methods utilized Transformer structure not only in the feature fusion stage but also as a backbone of feature extraction stage, such as SwinTrack(Lin et al., 2022). Since the feature extraction and fusion process are separated, the feature of search area could not perceive the target information during the feature extraction, so the two-stage Transformer tracking models tend to overlook target-related detail information.

Subsequently, researchers developed single-stage Transformer-based object tracking approaches, which integrated the feature extraction and fusion phases. Cui et al. proposed Mixformer(Cui et al., 2022), which introduced a hybrid attention module to extract discriminative target-related feature and facilitate information exchange between the template and search area. A unified framework OSTrack(Ye et al., 2022) was proposed for the feature extraction and relationship modeling, which contained an candidate elimination module to discard irrelevant spatial features, thereby improving the inference speed of model. (Song et al., 2023) introduced a compact Transformer-based tracker (CTTrack), which employed an asymmetric hybrid attention mechanism to strengthen feature. ProContEXT(Lan et al., 2023) synergistically utilized the spatial and temporal contextual information, and (Wu et al., 2023) introduced a masked auto-encoder into the Transformer-based object tracking. LiteTrack(Wei et al., 2024) aimed to enhance tracking speed and made the Transformer-based tracking model applicable to resource-constrained devices. (Zhang et al., 2023) leveraged semantic information from language to compensate for the instability of visual information, and proposed a language-based evaluation tracking method for selecting high-quality target samples. (Song et al., 2023) added a lightweight masked encoder to the single-stage tracking framework, guiding the encoder to capture invariant features. (Gao et al., 2023) proposed a generalized relationship model by adaptively selecting template and search area tokens for more flexible interactions. (Kang et al., 2023)designed a bridging module which integrated the high-level semantic information from deep features into the shallow high-resolution features to ensure high accuracy while maintaining fast operation on different devices.

The single-stage Transformer-based object tracking models often suffer from insufficient feature extraction capability, leading to potential confusion between the target and background during the

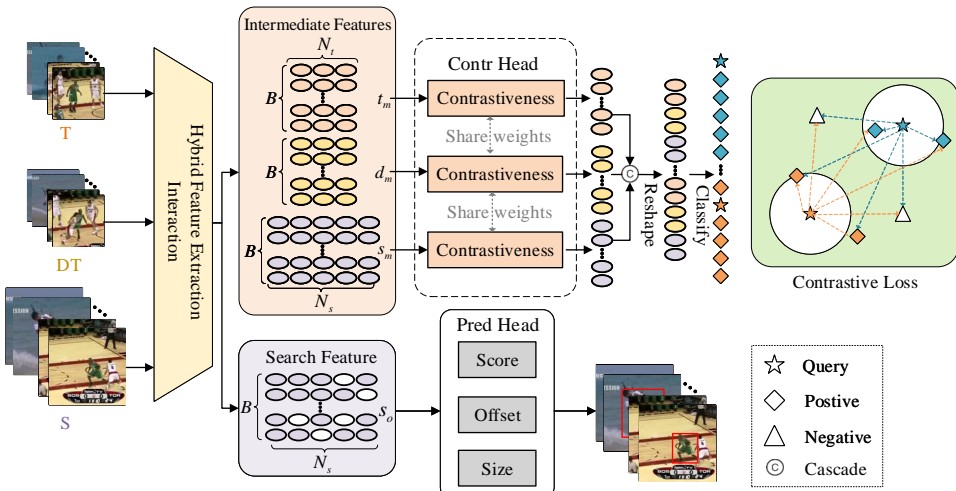

Figure 1: The training framework of HCTrack. $B$ (batch) represents the number of sample pairs, $N_t$ denotes the number of template feature tokens and dynamic template feature tokens, and $N_s$ indicates the number of search area feature tokens.

interaction of template and search area.Therefore, we introduce contrastive learning to mitigate this issue in this paper.

## 2.3 CONTRASTIVE LEARNING

Contrastive learning(Oord et al., 2018; Khosla et al., 2020; Seo et al., 2024) is to learn feature representation of samples by comparing the similarities or differences among different samples. Recently, contrastive learning has also been applied in the object tracking.(Wu et al., 2021) proposed a progressive unsupervised object tracking learning framework, which distinguished between target and background regions in video sequences through contrastive learning. (Pi et al., 2022) proposed an instance-aware module to enhance the separability among instances and the compactness within instances through using contrastive learning mechanisms. (Zeng et al., 2023) employed contrastive learning to learn discriminative representation of targets in the drone object tracking. (Bhat et al., 2019) applied contrastive learning to improve object-background discriminability in visual tracking, enabling the tracking model to learn a robust object-specific appearance model online.

However, the number of positive samples is relatively small and all of them are based on CNN in the aforementioned methods. In this paper, we introduce contrastive learning into the Transformer-based object tracking method, which leverages contrastive learning in a simple yet effective manner to enhance the discriminability of model, thereby improving tracking accuracy without compromising inference speed.

## 3 PROPOSED METHOD

### 3.1 OVERVIEW

To enhance the ability of model to distinguish between the target and the background, we propose a Hybrid Contrastive Transformer Tracker (HCTrack) in this paper, as illustrated in Figure 1. HCTrack consists of a hybrid feature extraction and interaction module (HFEI), a contrastive head (Contr Head), and a prediction head (Pred Head) including score, bias and scale head.

As shown in Figure 1 , the input of HCTrack comprises multiple sample pairs from video sequences. Each sample pair consists of a template image (T), a dynamic template image (DT) and a search frame image (S), where the sample pairs from the same video sequence are considered positive samples, while those from different video sequences are treated as negative samples. Then, the middle features of the template image, the dynamic template image and the search frame image are ob-

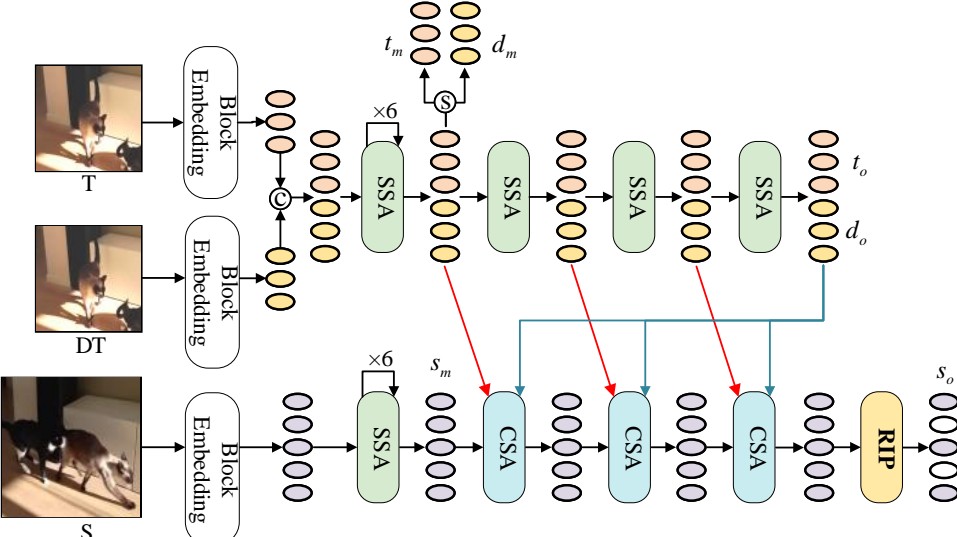

Figure 2: The structure of hybrid feature interaction module in HCTrack. S represents the feature separation operation. C represents the cascade operation.

tained through a hybrid feature extraction and interaction module. Subsequently, these intermediate features are respectively fed into three contrastive heads sharing weights for feature mapping, thus generating feature vectors of the template, dynamic template, and search area. After these feature vectors from the same video sequence are cascaded and reshaped, they are divided into positive and negative sample feature vectors to calculate the contrastive learning loss function. Meanwhile, the search area feature is fed into the score head, offset head and scale head to predict the bounding box of target, with the prediction process being supervised by classification and regression loss functions. It is worth noting that the contrastive head is only used during training and therefore does not affect the inference speed of model.

## 3.2 HYBRID FEATURE EXTRACTION AND INTERACTION MODULE

In HCTrack, we construct a hybrid feature extraction and interaction module to extract the features of templates and search frames and realize their information interaction. Figure 2 shows the structure of HFEI, which contains several semantic self-association (SSA) modules, several cross-layer semantic association (CSA) modules, and a redundant information pruning (RIP) module. SSA consists of a multi-head self-attention, a skip connection and a multilayer perceptron.

Firstly, the three types of images pass through a block embedding layer respectively to attain template feature token, dynamic template feature token, and search feature token, as shown in Figure 2. Then, the template feature token and dynamic template feature token are concatenated and processed by nine SSA modules for the template feature extraction, where the output of the sixth SSA module is separated as the middle template feature token $t_m$, and dynamic template feature token $d_m$. The final template feature token $t_o$ and dynamic template feature token $d_o$ obtained by last SSA module are used for the feature extraction of search area.

For the feature extraction of the search area in Figure 2, the middle search feature token $s_m$ is attained by six SSA modules before the feature interaction. Subsequently, the middle search feature token $s_m$, the current template feature token $t_m$, the current dynamic template feature token $d_m$, the final template feature token $t_o$, and the final dynamic template token $d_o$ are fed into a CSA module to achieve the interaction between the template and the search area. This process is repeated three times to produce the final search area feature $s_o$.

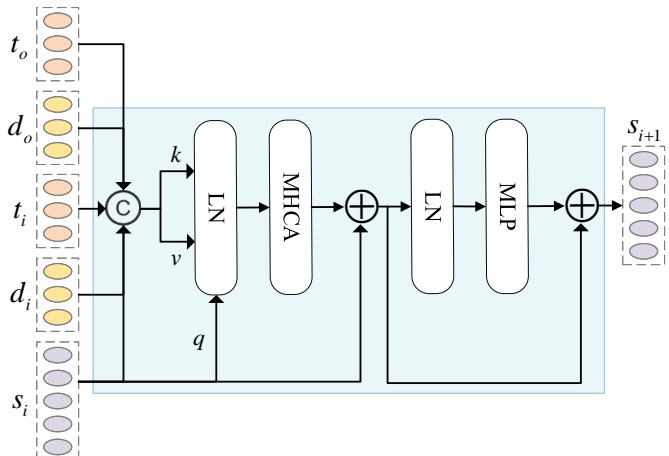

Figure 3: The structure of cross-layer semantic association module

### 3.3 CROSS-LAYER SEMANTIC ASSOCIATION MODULE

Most existing methods only utilized the current template features or the deep layer template features to guide the model in extracting target information from the search features, but the appearance of target in the search area is likely to differ significantly from that in the template, making it difficult to accurately locate the changing target by relying on single-layer template features solely. To better extract target semantic information from the search area, we design a Cross-layer Semantic Association (CSA) Module, which is applied to the deep layers of the search area branch in HFEI module. The structure of this module is shown in Figure 3, and its relationship with other modules in the HCTrack model can be referred to in Figure 1. Figure 3 shows the structure of CSA, which is composed of Multi-head Cross-Attention (MHCA), Layer Normalization (LN), and Multi-Layer Perceptron (MLP). For the MHCA operation, the current search feature token $s_i$ is first mapped to the query vector $q$ . Then, the search area feature token $s_i$ ,the final template feature token $t_o$ ,the final dynamic template token $d_o$ , the current layer template feature token $t_i$ ,and the current layer dynamic template token $d_i$ are concatenated together to generate the key vector $k$ and value vector $v$ for MHCA by key mapping and value mapping respectively.

### 3.4 REDUNDANT INFORMATION PRUNING MODULE

In order to further highlight the foreground target, we present a redundant information pruning (RIP) module to remove redundant background information. It is placed in the back of last CSA in HFEI module to prune the search area feature tokens acquired by the final layer of CSA module. The energy of each search token is first calculated, and then the top tokens with the most significant energy are retained.

**Energy Calculation**. Considering that the template includes some background information; we only retain the core template features and the attention weights for each search token to ensure the accuracy of pruning. Specifically, the similarity matrix between the search feature $s_i$ and the corresponding token in the central area of the template is averaged pooled to obtain the energy $e_i$ for each search token ($1 \leqslant i \leqslant N_s$ , where $N_s$ is the number of tokens in the search area).

**Pruning**. Afterwards, a proportion of the total energy $\theta$ of all search feature tokens is taken as an energy threshold, where the proportion is defined as the energy retention ratio $\rho$ ($0 < \rho < 1$). Next, the tokens are arranged in descending order according to their energy. Finally, the top $m$ tokens tokens with the highest energy that is greater than the energy threshold are retained, and the remaining tokens are discarded. The pruning calculation process after sorting is formularized as follows:

$$\theta = \sum_{i=1}^{N_s} e_i, \tag{1}$$

$$\underset{m}{\operatorname{argmin}} \left( \sum_{i=1}^{m} e_i \right) \geqslant \rho \cdot \theta. \tag{2}$$

The redundant information pruning module prunes the spatial features of the search area according to the similarity between the template center area and the search area. The smaller the similarity, the more redundant the corresponding position. In addition, the background information in the search area is adaptively discarded according to the complexity of the image to reduce the model's excessive attention to the background area, thereby improving the discrimination ability of the MCTrack model.

RIP module prunes the spatial features of the search area according to the similarity between the template center area and the search area. The smaller the similarity, the less possibility the corresponding features including the target information. In addition, the background in the search area is adaptively discarded according to the scene complexity, which can reduce the excessive attention to the background area, thereby improving the discrimination ability of HCTrack.

### 3.5 Contrastive Learning Module

Contrastive learning mainly mines the similarity among samples, which brings the same sample and its augmented data closer in the embedding space, and pushes the different samples farther apart, thereby enhancing the discriminability among samples. Therefore, we combine the contrastive learning with the Transformer-based object tracking to assist in training the model. Specifically, the contrastive learning strategy in this paper involves three key components: construction of positive and negative samples, contrastive head, and contrastive loss function.

**Construction of positive and negative samples**. In general, the positive samples are from the same image through data augmentation, and negative samples come from different images in self-supervised contrastive learning method. However, applying this approach directly to object tracking may ignore the temporal characteristics of video, potentially resulting in samples from the same video being treated as negative samples. To address this issue, we adopts a different way that the template images, the dynamic template images, and the search frame images from the same video sequence are treated as positive samples. These images contain the same target with richer variations, compared to the positive samples generated through data augmentation, which makes the model more robust. Meanwhile, the negative samples are drawn from other video sequences, which prevent the samples containing the same target from being misclassified as negative samples.

**Contrastive head**. HCTrack employs three contrastive heads with shared weights, which process the search area feature token $s_m$, the template feature token $t_m$ and the dynamic template feature token $d_m$ respectively, as shown in Figure 1. The contrastive heads are composed of region of interest pooling, a 3×3 convolution, a Rectified Linear Unit (ReLU), a 1×1 convolution, and batch normalization. Additionally, the output template feature vector, dynamic template feature vector, and search area feature vector are concatenated and reorganized. The feature vectors from the same video are grouped together as positive samples, while feature vectors from different videos are classified as negative samples. The positive and negative samples are then used to compute the contrastive learning loss function.

**Contrastive loss function**. In this paper, we give an improved contrastive learning loss function based on Info Noise Contrastive Estimation (InfoNCE) (Oord et al., 2018). The InfoNCE loss function is as follows:

$$L_{NCE} = -\log \frac{\exp\left(\frac{x \cdot k^+}{\tau}\right)}{\sum_{i=0}^{N} \exp\left(\frac{x \cdot k_i}{\tau}\right)}, \tag{3}$$

where $x$ represents the original sample, $k^+$ represents the only positive sample, $k^i$ represents any sample, $N$ represents the total number of samples, and $\tau$ is the temperature coefficient. In the formula (9), the numerator represents the similarity between the original sample and the positive sample, and the denominator represents the similarity between the original sample and all samples. This loss function learns the relationship among samples by maximizing the similarity of positive samples and minimizing the similarity with negative samples.

In InfoNCE loss function, the positive sample is unique, which may lead to insufficient coverage of the variations of target during training due to the limited number of positive samples. To address this issue, we propose an improvement to InfoNCE by increasing the number of positive samples. This enhancement promotes HCTrack to better learn the feature representation of the same target, thereby improving tracking accuracy. The improved InfoNCE loss function is as follows:

$$L_{cl} = -\log \frac{\sum_{i=0}^{n_{cl}^+} \exp\left(x\frac{k_i^+}{\tau}\right)}{\sum_{i=0}^{n_{cl}^+} \exp\left(x\frac{k_i^+}{\tau}\right) + \sum_{i=0}^{n_{cl}^-} \exp\left(x\frac{k_i^-}{\tau}\right)}, \qquad (4)$$

where, $n_{cl}^+$ is the number of positive samples, $n_{cl}^-$ is the number of negative samples, $k_i^+$ is the positive sample, and $k_i^-$ is the negative sample.

During training, the similarity among targets within the same video sequence and the dissimilarity between the target and other objects are strengthened gradually by minimizing the improved InfoNCE loss function. This process can help HCTrackaccurately capture the target information in the search area.

### 3.6 Loss Function

HCTrack utilizes a scoring head, a bias head, and a scale head for the bounding box prediction, which predicts the probability score of the central position of target, the discretization error, and the scale of target respectively. Additionally, a contrastive head is employed to compute the contrastive loss function.

Therefore, the total loss function of HCTrack consists of the contrastive loss function $L_{cl}$, the weighted focal loss classification $L_{cls}$, the L1 loss function $L_{1reg}$, and the GIOU loss function $L_{giou}$, which is as follows:

$$L = \lambda_{cl} L_{cl} + \lambda_{cls} L_{cls} + \lambda_{L_1 reg} L_{1reg} + \lambda_{giou} L_{giou}, \qquad (5)$$

## 4 Experiments

### 4.1 Implement Details

**Model**. HCTrack utilizes the first 9 encoder layers of ViT-Base(Dosovitskiy, 2020) as the backbone network and is initialized by using MAE(He et al., 2022) pre-trained parameters. The search region image size is 256×256, and after passing through the patch embedding layer, the token length in the backbone network is 256 with a channel dimension of 768. The template image size is 128×128, and after passing through the patch embedding layer, the token length in the backbone network is 64 with a channel dimension of 768. The redundancy pruning module retains 90% of the energy.

**Training**. For the tests on GOT-10k and TrackingNet datasets, the training datasets of HCTrack are different. To obtain the tracking result on GOT-10k test set GOT-10k dataset is only used for training according to GOT-10k protocol. For TrackingNet test set, a combination of COCO2017, TrackingNet, LaSOT, and GOT-10k datasets is used for training, which is consistent with most methods. The preprocessing process of the training data is same as that of OSTrack(Ye et al., 2022), including image cropping and data augmentation. The model is performed for 300 epochs, where $6 \times 10^4$ image pairs are processed in each epoch. Every image pair includes a static template image, a dynamic template image, and a search image. In addition, ADAM optimizer with a weight decay of $1 \times 10^{-4}$ is used, and the learning rate is initially set to $2 \times 10^{-4}$ and dropped to $2 \times 10^{-5}$ at 240 epochs.

**Inference.** At the beginning of testing, the first frame is set as both the initial static template and the dynamic template. The dynamic template is updated with the image that has the highest confidence score in an interval of 200 frames, where the confidence of each image is determined by the value output by the score head.

Table 1: The performance comparison of HCTrack with other methods on GOT-10k and Track-ingNet datasets. The models are named according to the format 'method name_resolution'. The best results are indicated in bold, and the second-best results are underlined. '-' represents that the information is not given.

| Model | Source | FPS | GOT-10k | | | TrackingNet | | |
|---|---|---|---|---|---|---|---|---|
| | | | AO(%) | $SR_0.5$(%) | $SR_0.75$(%) | AUC(%) | Pnorm (%) | P(%) |
| SiamFC_256 (Bertinetto et al., 2016) | ECCV16 | 86 | 34.8 | 35.3 | 9.8 | 57.1 | 66.3 | 53.3 |
| SiamRPN++_256 (Li et al., 2019a) | CVPR19 | 35 | 51.7 | 61.6 | 32.5 | 73.3 | 80.0 | 69.4 |
| DiMP (Dosovitskiy, 2020) | ICCV19 | 43 | 61.1 | 71.7 | 49.2 | 74.0 | 80.1 | 68.7 |
| SiamFC++ (Xu et al., 2020) | AAAI20 | 90 | 59.5 | 69.5 | 47.9 | 75.4 | 80.0 | 70.5 |
| Ocean (Zhang et al., 2020) | ECCV20 | - | 61.1 | 72.1 | 47.3 | - | - | - |
| STMTrack (Fu et al., 2021b) | CVPR21 | 28.6 | 64.2 | 73.7 | 57.5 | 80.3 | 85.1 | 76.7 |
| STARK (Yan et al., 2021) | ICCV21 | 31.7 | 68.8 | 78.1 | 64.1 | 82.0 | 86.9 | - |
| TransT_256 (Chen et al., 2021) | CVPR21 | 50 | 67.1 | 76.8 | 60.9 | 81.4 | 86.7 | 80.3 |
| SimTrack (Chen et al., 2022) | ECCV22 | - | 69.8 | 78.8 | 66.0 | 83.4 | 87.4 | - |
| Mixformer (Cui et al., 2022) | CVPR22 | 31 | 71.2 | 79.9 | 65.8 | 83.9 | **88.9** | 83.1 |
| OSTrack_256 (Song et al., 2023) | ECCV22 | 105 | 71.0 | 80.4 | 68.2 | 83.1 | 87.8 | 82.0 |
| SwinTrack_224 (Lin et al., 2022) | NIPS22 | 96 | 71.3 | 81.9 | 64.5 | 81.1 | - | 78.4 |
| CTTrack_320 (Lan et al., 2023) | AAAI2023 | 40 | 71.3 | 80.7 | 70.3 | 82.5 | 87.1 | 80.3 |
| VideoTrack_256 (Xie et al., 2023) | CVPR23 | - | 72.9 | 81.9 | 69.8 | 83.8 | 88.7 | 83.1 |
| TATrack_B224 (Zheng et al., 2023) | CVPR23 | 25 | 73.0 | 83.3 | 68.5 | 83.5 | 88.3 | 81.8 |
| GRM_256 (Gao et al., 2023) | CVPR23 | 45 | 73.4 | 82.9 | 70.4 | **84.0** | 88.7 | **83.3** |
| LiteTrack_256 (Wei et al., 2024) | ICRA24 | 170 | 72.2 | 82.3 | 69.3 | 82.4 | 87.3 | 80.4 |
| HCTrack_256 (ours) | | 163 | **73.6** | **84.2** | 69.7 | 82.6 | 87.6 | 80.5 |

## 4.2 PERFORMANCE COMPARISON

To measure the performance of the proposed HCTrack, we compare it with the current mainstream target tracking algorithms on GOT-10k and TrackingNet datasets, including SiamFC, SiamRPN++, DiMP, SiamFC++, Ocean, STMTrack, STARK, TransT, SimTrack, Mixformer, OSTrack, Swin-Track, CTTrack, VideoTrack, TATrack, GRM and LiteTrack. Table 1 shows the experimental results of HCTrack with other state-of-the-art models on GOT-10k and TrackingNet datasets.

**GOT-10k dataset:** It can be seen from Table 1 that the AO and SR0.75 scores of HCTrack is highest than other object tracking methods, which demonstrates the favorable performance of HCTrack. Meanwhile, FPS of HCTrack is higher than most tracking models in Table 1. It indicates that the introduction of the contrastive learning strategy can effectively improve tracking accuracy of the model under the condition of fast tracking.

**TrackingNet dataset:** It is observed that HCTrack is superior to other models except OSTrack in the tracking accuracy, as shown in Table 1. Although AUC of HCTrack is lower than that of SimTrack, Mixformer, OSTrack, VideoTrack, TATrack, and GRM, FPS of HCTrack is far higher than them. Therefore, our proposed HCTrack strikes a better balance between the tracking accuracy and speed.

## 4.3 ABLATION STUDY

To validate the effectiveness of each module in HCTrack, we conducted a series of ablation experiments on GOT-10k dataset. Table 2 shows the tracking results of different modules and their combinations in HCTrack, where the template features of the current layer is not contained in the input features of CSA module if CSA module is not selected in Table 2.

In Table 2, the baseline model without CSA, RIP, contrasting learning (CL) strategy and dynamic template update (DTU) mechanism, achieves AO of 71.8%. After incorporating RIP module into the baseline model, AO increases to 72.4%, which suggests that RIP module can effectively increase to the focus on the targets during tracking by eliminating redundant background information. When CSA module is added to the baseline model, AO increases to 72.5%, as shown in Table 2. This

Table 2: The effectiveness analysis of different modules in HCTrack.

| CL | DTU | CSA | RIP | AO(%) | SR0.5(%) | SR0.75(%) |
|----|-----|-----|-----|-------|----------|-----------|
| - | - | - | - | 71.8 | 81.9 | 68.3 |
| - | - | - | ✓ | 72.4 | 82.3 | 68.7 |
| - | - | ✓ | - | 72.5 | 82.7 | 69.4 |
| ✓ | - | - | - | 73.1 | 83.6 | 69.0 |
| ✓ | ✓ | - | - | 73.4 | 84.1 | 70.1 |
| ✓ | ✓ | ✓ | ✓ | **73.6** | **84.2** | 69.7 |

indicates that leveraging multi-layered template semantic information can enhance the ability of model to perceive and locate the targets.

In addition, after integrating the contrastive learning strategy into the baseline model, AO increases by 1.3% in Table 2. It suggests that our designed contrastive learning strategy can significantly improve the tracking accuracy of targets through learning the differences between positive and negative samples during training. On this basis, if the dynamic template update mechanism is also utilized to extend the positive samples in the contrastive learning of HCTrack, AO of the model increases to 73.4%, which further confirms the effectiveness of the contrastive learning strategy. Finally, the organic combination of CSA module, RIP module, contrasting learning strategy and dynamic template update mechanism makes sure the best performance of HCTrack in Table 2.

## 5 CONCLUSION

In this paper, we present a Hybrid Contrastive Transformer Tracker (HCTrack) to address the challenges of confusion of targets and backgrounds in the Transformer-based object tracking. By integrating contrastive learning with a Transformer-based tracking framework, HCTrack effectively enhances the ability to differentiate between targets and backgrounds, thereby improving tracking accuracy. Furthermore, a hybrid feature interaction module is constructed to realize comprehensive information exchange between template and search regions. In addition, we design a redundant information pruning module to reduce background interference by adaptively eliminating target-irrelevant redundant features based on global scene information. Experimental results on the GOT-10k and TrackingNet datasets demonstrate that HCTrack achieves competitive performance in terms of tracking accuracy, while maintaining fast tracking speed.

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

# A APPENDIX

## A.1 STRATEGY ANALYSIS

To validate the structure rationality of our proposed different modules in this paper, a series of experiments on GOT-10k dataset are conducted.

**(1) The structure analysis of CSA**

In this paper, we design a CSA module to extract rich template characteristics by associating the search region features with multi-level template features. The associated template features in CSA module can be selected in different ways, as shown in Figure 4. Table 3 shows the experimental results of the model only containing CSA with different association modes corresponding to Figure 4. The comparison of the experimental results corresponding to Figure4(a-c) in Table 3 shows that the template feature of last layer is very important for improving the tracking accuracy. When only the last layer of template features is used for CSA module shown in Figure4(a) , AO, $SR_{0.5}$, and $SR_{0.75}$ reach 71.8%, 81.9%, and 68.3% respectively in Table 3. In contrast, if the template features of current layer or penultimate layer are used as shown in Figure4(b) and (c), these metrics of Table 3 decrease. This shows that deeper template features (from the last layer) can more effectively describe the target information in the template than the shallower template features (from the current layer or the penultimate layer), thereby better guiding the model in extracting target information from the search area.

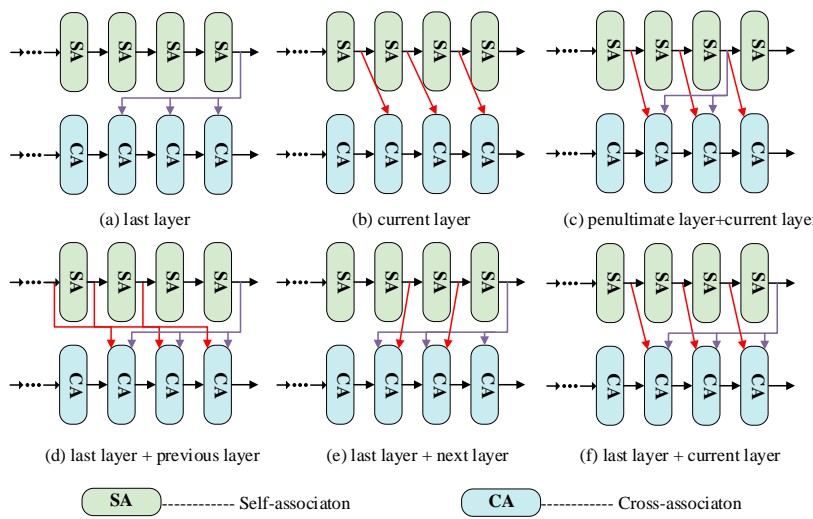

Figure 4: Different template input structures of the cross-layer semantic association module.

Table 3: The influence of different association modes in CSA module.

| Input template features | AO(%) | SR0.5(%) | SR0.75(%) |
|---|---|---|---|
| Figure 4 (a) | 71.8 | 81.9 | 68.3 |
| Figure 4 (b) | 70.3 | 80.4 | 66.4 |
| Figure 4 (c) | 71.4 | 80.9 | 68.0 |
| Figure 4 (d) | 71.6 | 81.3 | 67.8 |
| Figure 4 (e) | 72.0 | 81.7 | 68.7 |
| Figure 4(f) (CSA) | **72.5** | **82.4** | **69.3** |

Subsequently, under the premise of using the template feature of the last layerin CSA module, the template features of the previous layer, next layer, and current layer are added into CSA module respectively, as shown in Figure4(d-f). The corresponding tracking accuracies in Table 4 indicates that the association mode of Figure4(f), that is "last layer+current layer", has the best performance. It manifests that the template feature of current layer is more suitable for guiding the extraction of target information in the search area than the previous and next layers, resulting in better accuracy performance.

**(2) The deployment analysis of RIP**

The deployment of RIP module in different encoding layers of the backbone network can have different influenct on model performance. Table 4 shows the tracking accuracy of HCTrack when RIP module is deployed in different positions of backbone network. The module was placed in the seventh, eighth, and ninth layers of the backbone network respectively, and the corresponding AO gradually improves, reaching 72.2%, 73.0%, and 73.6%, respectively. This indicates that the deployment of RIP module on deep layer of backbone can more accurately remove the target-irrelevant backgrounds, thereby improving the tracking accuracy of the model.

**(3) The strategy analysis of positive sample selection**

HCTrack samples multiple pairs of instances from the same video sequence as the positive samples, and updates dynamic template to increase the number of positive samples. The different number of positive samples may affect the model performance. Table 5 shows the experimental results of HCTrack with different number of positive samples. When the number of positive sample pairs in the same video sequence is 1, 2, 3, and 4, the corresponding AO values are 72.5%, 73.3%, 73.6%, and 73.1%, respectively, as shown in Table 5, where the performance of model with three positive sample pairs achieves best. This indicates that the richness of positive samples is insufficient if the

Table 4: The influence of different deployment of RIP module.

| Deployment location | AO(%) | SR0.5(%) | SR0.75(%) |
|---|---|---|---|
| Layer 7 | 72.2 | 82.6 | 68.3 |
| Layer 8 | 73.0 | 83.6 | 69.5 |
| Layer 9 | 73.6 | 84.2 | 69.7 |

Table 5: The influence of different number of positive samples in HCTrack.

| Number of sample pairs in the same video | Whether to include dynamic templates | AO(%) | SR0.5(%) | SR0.75(%) |
|---|---|---|---|---|
| 1 | ✓ | 72.5 | 83.2 | 68.3 |
| 2 | ✓ | 73.3 | 83.6 | 70.2 |
| 3 | | 73.3 | 84.1 | 70.1 |
| 3 | ✓ | **73.6** | **84.2** | **69.7** |
| 4 | ✓ | 73.1 | 83.7 | 70.2 |

number of positive samples in the same video is small, making it difficult to accurately learn the core feature of the same target in different scenes. Conversely, when the number of positive samples is too large, the proportion of positive samples relative to the total number of samples becomes too high, resulting in relatively fewer negative samples and making the model difficult to learn the differences between different types of targets.

Additionally, under the premise that three positive sample pairs is utilized, we analysis the impact of dynamic templates in the positive samples on the model performance. After the dynamic template is removed, AO decreases from 73.6% to 73.3%, which indicates that setting dynamic templates as positive samples can increase the number of positive samples, guiding the model to better learn the relationship between positive and negative samples.

**(4) The feature selection analysis of contrast head**

In HCTrack, the features of the template, the dynamic template and the search area, are used for the contrastive learning features and fed into the contrastive head. Selecting the features from different layers of HFEI module as the inputs of the contrastive head has different influence on the tracking performance of HCTrack. Table 6 shows the experimental results of HCTrack when the features of different layers in HFEI module are used as the inputs of the contrastive head. With the increase

Table 6: The influence of different input feature of the contrast head in MCTrack.

| Number of feature layers | 1 | 2 | 3 | 4 | 5 | 6 | 7 | 8 | 9 |
|---|---|---|---|---|---|---|---|---|---|
| AO(%) | 68.5 | 70.1 | 69.6 | 71.7 | 73.2 | **73.6** | 71.5 | 70.4 | 70.2 |
| SR0.5(%) | 78.4 | 80.0 | 79.6 | 81.0 | 83.1 | **84.2** | 81.8 | 80.2 | 80.0 |
| SR0.75(%) | 63.7 | 66.0 | 65.2 | 68.0 | 71.4 | **69.7** | 67.4 | 67.1 | 66.7 |

of the layer number of the feature used for the contrastive head, AO score goes up and then down, where the tracking accuracy peaks when using features extracted from the 6th layer as input of the contrastive head, achieving an AO of 73.6%. This suggests that the shallow features from earlier layers lack the representational capacity to effectively distinguish between targets and backgrounds, resulting in poorer tracking accuracy. On the other hand, since the deeper search features from CSA module have contained the template information, the diffence between the positive samples and the negative samples becomes small, resulting in performance degradation.

A.2 Parameter analysis

During training HCTrack, a contrastive loss function was introduced. The varying weights $\lambda_{cl}$ of this loss function have different impacts on the performance of HCTrack. When only dynamic template updating mechanism and contrast learning are used, the experimental results of the model

on GOT-10k dataset under different contrast loss function weights are presented in Table. 7. It can be observed that the model achieves the best performance when $\lambda_{cl} = 0.1$ . However, AO gradually decreases with $\lambda_{cl}$ rising from 0.1 to 0.9. As the proportion of the contrastive learning loss function increases, the corresponding proportion of regression and classification loss function decreases, leading to a gradual decline in accuracy. It suggests that the regression and classification are the key to the tracking task, and the contrastive learning loss function can serves as an auxiliary component to helpg the models achieve optimal performance.

Table 7: The weight analysis of comparative loss function in HCTrack.

| $\lambda_{cl}$ | 0.0 | **0.1** | 0.2 | 0.3 | 0.4 | 0.5 | 0.6 | 0.7 | 0.8 | 0.9 |
|---|---|---|---|---|---|---|---|---|---|---|
| AO(%) | 73.1 | **73.4** | 72.7 | 69.7 | 70.6 | 71.4 | 70.5 | 69.1 | 69.3 | 69.1 |
| SR0.5(%) | 82.6 | **84.1** | 82.9 | 79.5 | 80.2 | 81.2 | 80.3 | 78.8 | 79.1 | 78.6 |
| SR0.75(%) | 69.4 | **70.1** | 69.7 | 66.5 | 66.4 | 67.7 | 66.8 | 65.3 | 65.7 | 64.8 |

