# OpenReview forum: "Hybrid Contrastive Transformer for Visual Tracking"
_ICLR.cc/2025/Conference — Submitted to ICLR 2025_

### Official Review · Reviewer_w8t9 · 2024-10-18

**Soundness:** 3
**Presentation:** 2
**Contribution:** 2
**Rating:** 3
**Confidence:** 5

**Summary:**

This paper introduces the Hybrid Contrastive Transformer Tracker (HCTrack), which uses contrastive learning to enhance target-background differentiation. The author proposes a hybrid feature interaction module for better information exchange between template and search regions, along with a pruning module to eliminate redundant background elements. The proposed HCTrack achieves good results on some datasets, such as the GOT-10k. Some benchmarks, such as TrackingNet, are fair.

**Strengths:**

1. The motivation to apply contrastive learning to the visual tracking area is good and makes sense. The discrimination between template and search area is vital for robust visual tracking.

2. The author has conducted some work and effort to apply contrastive learning to visual tracking, such as a hierarchical feature interaction module and contrastive loss on the prediction head.

3. The experiments seem to provide some useful and reasonable conclusions.

**Weaknesses:**

1. The overall method does not surprise much. The hierarchical feature interaction for the transformer is straightforward and naive. The contrastive loss on prediction heads also lacks enough insights for a top conference.  The method seems to be a reconstruction of the attention layer for feature fusion.

2. This paper's biggest drawback is the experiments. The experiments and ablations are not enough to prove the overall methods. Only two datasets are used: got10k and Trackingnet. The performance in the Trackingnet benchmark is not good.

3. The presentation of this work is not academic. For example， “S represents the feature
separation operation. C represents the cascade operation.” In another sentence, "a search
frame image (S)."

**Questions:**

Please see the weakness.  The most concerning aspect is that the method is straightforward and lacks in-depth insights. The experimental results are weak and not enough to prove the effectiveness. The overall presentation of this paper does not reach the standard of a top-tier conference.

---

### Official Review · Reviewer_gcrm · 2024-10-21

**Soundness:** 3
**Presentation:** 2
**Contribution:** 2
**Rating:** 6
**Confidence:** 4

**Summary:**

In this paper, the author tries to improve the feature extraction ability of trackers. They think the bottleneck of tracking is potential confusion between the target and background. Therefore, they introduce contrastive learning into the tracking field. Besides, to raise the performance of tracking, they propose a semantic self-association module and a cross-layer semantic association module. Those two modules can make full use of multi-level template features. Finally, a redundant information pruning module is built for pruning the redundant background
information.

**Strengths:**

1.  The author introduces a new contrastive loss into the transformer-based trackers to overcome the bottleneck of feature extraction capability.
2. To utilize the multi-level feature of tracking images, they propose a self- and cross-association module. The ablation studies prove the effectiveness of those two modules.
3. Comprehensive evaluations of HCTrack are conducted, including parameters analysis, architecture, and strategy analysis.

**Weaknesses:**

1. The improvement in tracking performance brought by contrastive learning and other proposed modules is small. According to Table 1, the performance of HCTrack is similar or even lower than the accuracy of previous works.
2. The author only uses two tracking benchmarks. The author should consider adding more data to validate the performance of HCTrack, such as LaSOT.
3. The qualitative comparison is lacking. The author should add some qualitative comparison behind the quantitative evaluations.
4. Some representative temporal tracking works should be analyzed in the related work section. For example, TrDimp [1] and TCTrack [2].
5. I hope the author can release the related code and resources.

[1] Wang N, Zhou W, Wang J, et al. Transformer meets tracker: Exploiting temporal context for robust visual tracking[C]//Proceedings of the IEEE/CVF conference on computer vision and pattern recognition. 2021: 1571-1580.
[2] Cao Z, Huang Z, Pan L, et al. TCTrack: Temporal contexts for aerial tracking[C]//Proceedings of the IEEE/CVF conference on computer vision and pattern recognition. 2022: 14798-14808.

**Questions:**

I want to know the baseline of this paper. Why does the author choose to build a new tracker rather than introducing contrastive learning into a previous tracker? In my opinion, developing a contrastive module for existing trackers may be a better choice.

**Details Of Ethics Concerns:**

I don't think there are severe ethical concerns

---

### Official Review · Reviewer_TZfB · 2024-10-31

**Soundness:** 2
**Presentation:** 3
**Contribution:** 1
**Rating:** 3
**Confidence:** 5

**Summary:**

The authors proposed to use contrastive learning in order to mitigate the ineffeciencies in transformer-based feature extraction.  They proposed a feature interaction module that allows target features from different levels of the extractor network to interact with search image features.  The authors also propose an improved InfoNCE and a Redundant information pruning (RIP) module.

**Strengths:**

- This paper is likely early to enquire the effect of contrastive learning in transformer based object trackers. Any notable findings in this area have the potential to drive meaningful advancements in transformer-based object tracking.
- The related works section is well written and the rest of the paper is easy to go through along with the well explanative figures.
- They have also proposed a new improved InfoNCE loss.
- Cross layer attention model architectures are well researched (in the appendix)

**Weaknesses:**

- Lacks novelty: contrastive loss in vision transformers, and contrastive loss on single object tracker are separately already explored in the field. Paper claims to mitigate the issues in transformer feature extractor specifically in tracker environments, by exploring contrastive learning options for transformers. However the contrastive loss application is not well ablated over different transformer architectures.
- Lacks experimental backing:
    - results are published only on two old datasets GOT-10k and TrackingNet, missing out other important benchmarks such as LaSOT. The Train split of LaSOT, however, is utilized for training HCTrack.  No explanation is provided for not testing the tracker on LaSOT. Including LaSOT results would ensure broader validation and address a key benchmark in tracking research
    - Results on TrackingNet are poor compared to other trackers. Authors reasoned out that  HCTrack is faster, however, authors should have explored impact of CL on a larger sized tracker, or on higher resolution input to prove that their model scales with more params/computations to achieve better scores.
    - No ablation test is provided on the improved InfoNCE when compared to the original InfoNCE from (Oord et al., 2018)

**Questions:**

- How well does the technique work for larger sized tracker models (which are comparable fast as the trackers that outperformed  HCTrack  on TrackingNet)?

---

### Official Review · Reviewer_Q88p · 2024-11-03

**Soundness:** 2
**Presentation:** 2
**Contribution:** 2
**Rating:** 3
**Confidence:** 5

**Summary:**

This paper develops a hybrid contrastive transformer for visual tracking. It contains semantic self-association and cross-layer semantic association modules to update the multi-level template features for more robustness. Besides, a redundant information pruning module is proposed to alleviate the influence of complex backgrounds on the target. The experiments are conducted on both GOT-10k and TrackingNet datasets, where a detailed ablation study is used to demonstrate the effectiveness of important designs.

**Strengths:**

1. This paper combines constrastive learning and transformer architecture into visual tracking. It helps differentiate the target from complex background.
2. The running speed of the proposed method is much higher than most compared methods. It shows the efficiency of HCTrack.

**Weaknesses:**

1. Contrastive learning is not new in visual tracking. When merged into the transformer architecture, it is essential to highlight the insights of the cross-attention module. Why does it work? In Sec. 3.3, I am not convinced that the proposed network is especially designed for visual tracking. It seems straight-forward to concat all the input features and fed into the attention module to compute the connections automatically. The authors should highlight why the architecture is designed.
2. In all the figures, I recommend that the authors can explain the meaning of variables for better understanding. Then the readers don't need to refer to the text.
3. There are a lot of hyper-parameters. In Sec. 4.1, the authors didn't describe how these parameters are set.
4. I have a big concern on the experiments. According to the GOT-10k leaderboard (http://got-10k.aitestunion.com/leaderboard), the performance of the proposed method is not even state-of-the-art. Even on Table 1, the improvement over selected compared methods are minor. I suggest that more recent trackers since 2022 can be compared, such as MixFormer, ARTrack, and SeqTrack.
5. Moreover, experiments on only two datasets are not comprehensive to show the superiority. For example, LaSOT and UAV related datasets can be added in the experiment section.
6. Another concern is on the ablation study. According to Table 2, it seems both CSA and RIP only has a slight improvement on AO and SR0.5, but decrease on SR0.75. It indicates that the proposed modules are not that effective. I suggest that authors can provide more explaination on this decrease.
7. How does this tracker deal with challenges in tracking, such as severa occlusion, and fast motion? The authors can show quantitative results on these challenges for a comprehensive evaluation.

**Questions:**

More compared methods and datasets are needed to enhance the experiment. Please explain the performance concern above.

---

### Meta-Review · Area_Chair_t5YE · 2024-12-21

**Metareview:**

Given the unanimous negative feedback from the reviewers and the absence of a response from the authors to address the concerns raised, it is concluded that the manuscript is not suitable for publication. Consequently, the decision is to reject the paper.

**Additional Comments On Reviewer Discussion:**

Given the unanimous negative feedback from the reviewers and the absence of a response from the authors to address the concerns raised, it is concluded that the manuscript is not suitable for publication. Consequently, the decision is to reject the paper.

---

### Decision · Program_Chairs · 2025-01-22

Reject